# Inositols Depletion and Resistance: Principal Mechanisms and Therapeutic Strategies

**DOI:** 10.3390/ijms22136796

**Published:** 2021-06-24

**Authors:** Elisa Lepore, Rosa Lauretta, Marta Bianchini, Marilda Mormando, Cherubino Di Lorenzo, Vittorio Unfer

**Affiliations:** 1R&D Department, Lo.Li. Pharma, 00156 Rome, Italy; e.lepore@lolipharma.it; 2Oncological Endocrinology Unit IRCCS Regina Elena National Cancer Institute, 00128 Rome, Italy; rosa.lauretta@ifo.gov.it (R.L.); marta.bianchini@ifo.gov.it (M.B.); marilda.mormando@ifo.gov.it (M.M.); 3Department of Medico-Surgical Sciences and Biotechnologies, La Sapienza University Polo Pontino, 04100 Latina, Italy; cherub@inwind.it; 4The Experts Group on Inositol in Basic and Clinical Research (EGOI), 00161 Rome, Italy; 5System Biology Group Lab, 00161 Rome, Italy

**Keywords:** myo-Inositol, d-chiro-Inositol, depletion, dietary supplementation

## Abstract

Inositols are natural molecules involved in several biochemical and metabolic functions in different organs and tissues. The term “inositols” refers to five natural stereoisomers, among which myo-Inositol (myo-Ins) is the most abundant one. Several mechanisms contribute to regulate cellular and tissue homeostasis of myo-Ins levels, including its endogenous synthesis and catabolism, transmembrane transport, intestinal adsorption and renal excretion. Alterations in these mechanisms can lead to a reduction of inositols levels, exposing patient to several pathological conditions, such as Polycystic Ovary Syndrome (PCOS), hypothyroidism, hormonal and metabolic imbalances, like weight gain, hyperinsulinemia, dyslipidemia, and metabolic syndrome. Indeed, myo-Ins is involved in different physiological processes as a key player in signal pathways, including reproductive, hormonal, and metabolic modulation. Genetic mutations in genes codifying for proteins of myo-Ins synthesis and transport, competitive processes with structurally similar molecules, and the administration of specific drugs that cause a central depletion of myo-Ins as a therapeutic outcome, can lead to a reduction of inositols levels. A deeper knowledge of the main mechanisms involved in cellular inositols depletion may add new insights for developing tailored therapeutic approaches and shaping the dosages and the route of administration, with the aim to develop efficacious and safe approaches counteracting inositols depletion-induced pathological events.

## 1. Introduction

Inositol and its phosphate derivatives are natural molecules involved in several biochemical and metabolic functions in different organs and tissues [1]. As part of membrane phospholipids, they are essential components of the cells, playing a crucial role as second messengers both in transduction pathways [2], including cellular growth, membrane biogenesis, signal transmission of hormones and neurotransmitters [3,4], and in several physiological processes such as reproductive, endocrine and metabolic pathways [5].

Notably, the term “inositols” refers to five natural stereoisomers, among which myo-Inositol(myo-Ins) is the most abundant one.

Several mechanisms contribute to maintain myo-Ins physiological levels, including food-dependent adsorption, biosynthesis and catabolism, intestinal and cellular uptake [6,7] and the administration of pharmacological therapies [8]. Imbalances of these mechanisms can alter inositol metabolism, causing a depletion of myo-Ins levels, which has been related to several chronic diseases, including metabolic syndrome, diabetes, polycystic ovary syndrome (PCOS) [9,10] and cancer, especially lung and breast cancer [11,12].

For instance, several studies confirmed that myo-Ins intracellular depletion is a common condition in both diabetic animal models and human subjects [13,14]: high glucose levels may reduce inositol availability by increasing its degradation and by inhibiting its biosynthesis and adsorption [7]. In addition, studies on patients with diabetes mellitus or chronic renal failure demonstrated a correlation with an altered inositol metabolism [6]. Previous studies revealed that a condition of insulin resistance is characterized by inositols imbalance also involving another stereoisomer, the d-chiro-Inositol (d-chiro-Ins) [15,16]. Physiologically, myo-Ins is converted into d-chiro-Ins through the activation of the insulin-dependent epimerase enzyme [17]. Therefore, myo-Ins depletion may also influence the downstream conversion into d-chiro-Ins, determining reduced levels of both stereoisomers [18]. Considering that the epimerization process only converts myo-Ins into d-chiro-Ins, without the inverse reaction, the administration of d-chiro-Ins only is not therapeutically beneficial, since it cannot restore myo-Ins levels. On the contrary, the supplementation of myo-Ins may recover the reduced levels of both stereoisomers, due to the physiological conversion. The concomitant administration of both stereoisomers leads to a noticeable therapeutic advantage: it directly provides a small amount of d-chiro-Insthat can be immediately functional to restore its altered levels, in addition to the physiological conversion starting from myo-Ins.

Consistent with this evidence, recent studies revealed that inositols administration restores glucose metabolism [18,19,20], insulin resistance and dyslipidemia in postmenopausal women with metabolic syndrome [21], in diabetic patients [22], and in pregnant women with gestational diabetes or at risk of developing the disease [23,24].

Besides metabolic alteration, myo-Ins depletion may also expose patients to several neuropathological and psychiatric conditions, including Alzheimer’s and Parkinson’s disease, amyotrophic lateral sclerosis and depression, suggesting a protective role for myo-Ins in different neurodegenerative and neurological disorders [25,26]. Previous studies reported that the administration of myo-Ins exhibited positive effects in recovering physiological intracellular levels in conditions such as obsessive–compulsive disorder and panic disorder [27], highlighting the connection between lower myo-Ins levels and the pathophysiology of depression [28,29].

Furthermore, inositol depletion may also influence fetus physiological development: several studies reported a relationship between lower levels of myo-Ins and the risk of craniofacial and neural tube defects (NTDs) [30,31]. Indeed, several clinical trials revealed that myo-Ins administration in pregnancy can be a useful preventive treatment promoting a trouble-free gestation [24].

Overall, a deeper knowledge of the main mechanisms involved in cellular inositol depletion may add new insights concerning the possibilities to supplement its levels, counteracting pathological events as a consequence of lower inositols levels. Among the downstream processes affected by myo-Ins depletion, the reduced conversion of myo-Ins into d-chiro-Insdetermines a consequent reduction of d-chiro-Inslevels, strengthening the importance of a combined supplementation.

The aim of this review is to gather principal mechanisms causing inositols depletion and resistance paving the way for tailored therapeutic strategies based on inositol supplementation.

## 2. Inositol Depletion Induced by Reduced Nutritional Availability

Inositol deficiency may arise from a variety of mechanisms, including a reduced dietary intake. Indeed, along with biosynthetic pathways, food-dependent intake is the primary source of inositols in mammals [17].

The human diet provides inositols, mostly myo-Ins, both from animal sources, in a free form or as inositol-containing phospholipids, and from plant foodstuffs as its phosphate derivative, namely InsP6 or phytic acid [32]. A high-fiber diet is one of the main sources of phytate, especially cereals and legumes, including oil seeds and nuts. They represent about 40% and 60% of total caloric intake for humans in developed and developing countries, respectively [32].

Interestingly, several studies have pointed out that the beneficial effects of a high-fiber diet are mainly due to the presence of inositol as a protective agent [7]. In particular, the positive effect is related to the presence of whole-grain cereal products: indeed, most phytate (approximately 80%) is located in the aleurone layer and the germ of the cereals (wheat, rice, etc.), while the endosperm is almost free of phytate [33,34].

A crucial point is the quantity of inositol that the human body can obtain from phytic acid. Dietary supplementation of inositols commonly used in clinical practice ranges from 2 to 4 g/daily [35,36], in line with the endogenous production of the human body (liver and kidney), which is up to 4 g/daily [37]. A balanced diet containing 5 g of phytic acid can provide as much inositol as can be obtained from nutritional supplementation, while a mixed Western diet provides about 1 g/daily, which is less than nutritional supplementation [38], contributing to reduced inositols levels.

Noticeably, myo-Ins and InsP6 exert plenty of positive effects on human health, including anti-diabetic, antioxidant, anti-inflammatory, and anticancer effects [12,39]. On the contrary, Western diets are rich in fat, sugar and poor in fibers, lacking components such as inositols and phytates, whose depletion was linked to the pathogenesis of chronic diseases such as metabolic syndrome, obesity, cardiovascular diseases, PCOS, diabetes, and cancer [7].

However, the nutritional effect of phytic acid has been widely discussed since the 1970s [7] due to the hypothetical “antinutritional” effect of InsP6 for its capacity to form insoluble salts, thus reducing minerals adsorption. These findings prompted the adoption of different strategies to eliminate phytate from food in order to counteract the unwarranted consequences related to the antinutritional effect of phytic acid. Therefore, processes aiming to reduce InsP6/myo-Ins content in several aliments, including soaking and/or extracting food or enhancing fermentation, created phytate hydrolysis products with weaker mineral-binding properties. Over the years, several studies have clarified this supposed “antinutritional” effect demonstrating that it occurs only when large quantities of InsP6 are consumed in combination with an unbalanced, oligo elements-poor diet [40,41]. Indeed, a consumption of 1–2 g/daily of InsP6 does not significantly affect the mineral status in humans [42].

Notably, recent studies revealed that the consumption of a high-refined diet is associated with an increased risk for cancer [43], while whole-grain cereal consumption exhibits a protective effect against mortality for inflammatory-related diseases [39]. Over-processed grains were stripped of germ and bran during the milling process, resulting in lower amounts of fiber and micronutrients, including inositols, and lower health benefits [44].

Additionally, the dietary assumption of inositol and phytate is very different among worldwide countries, with a great variability also reflecting diversity according to social and educational habits and origins. In line with this, two recent independent studies pointed out great differences depending on the age and gender of subjects in the Indian [45] and Egyptian [46] population as well as in many other countries [32].

In Western countries, daily intake of inositols and derivatives does not exceed 500–700 mg/day, while in Africa and Asia, a higher consumption is reported [7]. In Europe, InsP6 is far below daily doses; for instance, a recent study in Italy indicated a very lower range of phytic acid intake compared to the supposed daily dosage (from 219 to 293 mg/daily) [14,47]. In the USA and Canada, the average phytic acid intake is 538 mg/daily in adults, with relevant differences between males and females (608 mg vs. 512 mg/daily, respectively) [48], and 170–390 mg/daily in children [49]. On the contrary, in adult Asian immigrants to Canada consuming a vegetarian diet, the mean daily InsP6 intake increased [50]. Another study focused on the composition of the Korean diet, which is enriched with rice and other whole grains and sea vegetables, unlike the Western diet [51]. The authors reported a reduced risk of metabolic syndrome, obesity, hypertension, and hypertriglyceridemia among individuals taking a Korean diet [52]. Furthermore, another study evaluated the effects of a traditional Korean diet compared to a control diet (“eat as usual”) on cardiovascular risk factors in patients with diabetes or hypertension: positive effects were observed on body mass index, heart rate and glycemic control [53].

The resulting inositols deficiency due to different inositol and phytate contents according to dietary habits or to highly refined processes, may expose individuals to the risk of various diseases. Diets providing high amounts of fibers exhibit a protective role in the management and prevention of chronic diseases, including obesity, diabetes, PCOS, metabolic syndrome, cardiovascular diseases, and cancer [49,54,55,56,57]. On the contrary, the consumption of highly refined diets, principally based on pasta, rice, and bread, is frequently associated with an increased risk for diabetes and cancer [39,43,58].

The World Health Organization (WHO) included strategies of diet patterns in its global plan for preventing noncommunicable diseases [59], corroborating the crucial role of food and nutrients in determining the risk of certain diseases. A meta-analysis correlated the consumption of whole-grain intake to a reduced risk of noncommunicable diseases, such as stroke, cardiovascular diseases, cancer, and infections [44,60,61].

Overall, all the reported data corroborate the crucial role of inositols dietary availability, highlighting how the intake of certain types of nutrients positively influences human health.

## 3. Defects in Inositols Biosynthesis

Besides being adsorbed from diet, myo-Ins can be synthetized by organisms, and its biosynthetic pathways are strictly regulated and highly conserved throughout evolution [62].

Living organisms, from yeast to mammals, can synthetize up to 4–5 g of myo-Ins daily. In mammals, myo-Ins synthesis principally occurs in kidneys; however, most tissues are able to produce it endogenously [63,64], including liver, testis, and brain [65,66], starting from glucose uptake. In detail, inositol phosphate, which is the immediate precursor of free inositol, is synthesized de novo, starting from the internalization of glucose-6-phosphate (G6P). The latter is converted into inositol triphosphate (InsP3) by inositol-3-phosphate synthase (inositol synthase, INO1, or MIPS1), which is encoded by *ISNYA1* gene in mammals and expressed in all tissues, particularly in testis, heart, pancreas, ovary, and placenta [67]. The enzyme MIPS catalyzes the first step in the biosynthesis of all myo-Inositol-containing compounds [27] by converting glucose-6-phosphate to myo-Inositol-1-phosphate (MIP). Finally, InsP3 is dephosphorylated to free myo-Ins by Inositol monophosphatase-1 enzyme (IMPA-1 or IMPase) [3]. Free myo-Ins is also obtained by cyclic synthesis [68] and hydrolysis of phosphatidylinositol, in particular by recycling Inositol-1,4,5-trisphosphate (InsP3) and Inositol-1,4-bisphosphate (InsP2).

The biosynthetic pathway can be regulated at different levels and through several mechanisms, including epigenetic modifications of genes codifying for the involved enzymes. The epigenetic regulation consists of heritable phenotype changes to the genome that do not involve a change in the nucleotide sequence. In particular, epigenetic factors may modulate the expression of *ISNYA1* gene, codifying for inositol synthase in mammals (MIPS), leading to alternatively spliced isoforms, one of which may negatively modulate enzyme activity [69]. Additional findings suggested that DNA methylation pattern of *ISYNA1* gene is gender- and tissue-specific and negatively regulates the inositol biosynthetic pathway: significant alterations in methylation patterns during development could impact on inositol phosphate synthase expression in later life [27,69].

The *ISYNA1* gene is expressed in adult human tissues as well as in the placenta and yolk sac [67,70], leading to the hypothesis that genetic defects of this enzyme may result in low maternal and/or embryonic intracellular myo-Ins concentrations, predisposing one to NTD pathogenesis [71]. Indeed, inositol levels are crucial in the fetal phase, so much that its altered metabolism and levels can lead to pathological defects. High fetal myo-Ins concentrations in the cerebrospinal fluid, as well as in plasma and in amniotic fluid, have been correlated to the pathogenesis of Down Syndrome [72,73,74]. Indeed, gene products of the extra chromosome 21 can influence levels of certain metabolites, including myo-Ins. The gene *SLC5A3*, codifying for the inositol transporter SMIT1, is located on chromosome 21; therefore, the increased levels of myo-Ins can be associated with the upregulation of such transporter, even in the amniotic fluid, leading to an early detection of the syndrome [73]. On the contrary, a fetal deficiency of inositol can cause NTDs [75,76] and cranio-facial impairments [31]. In line with this, several studies reported crucial roles of inositol phosphate in the developement of the central nervous system (CNS), suggesting a protective role of inositol supplementation in reducing NTDs risk [24,30].

A positive regulator of *ISYNA1* transcription in mammalian cells is the glycogen synthase kinase 3 (GSK3), which also regulates several glycolytic pathways [77,78]. GSK3 is required for optimal *ISYNA1* activity, since a loss of its activity was related to a decreased activity of MIPS, resulting in myo-Ins depletion [79].

Furthermore, since GSK3 contributes to the regulation of glucose homeostasis, during the last few years, increasing attention has been focused on diabetes-associated changes in GSK3 and in particular on the association between low levels of inositols and altered glucose metabolism. Indeed, several studies confirmed the correlation between altered inositol metabolism and metabolic changes, including diabetes, obesity, and metabolic syndrome [14,64,80].

Inositol biosynthetic activity is highly relevant in the human body and the biosynthetic capability varies among different tissues, according to functional requirements. In particular, besides kidneys, the brain produces high levels of myo-Ins, which are strictly dependent on its biosynthesis and recycling, since inositol adsorption through the blood–brain barrier (BBB) is poorly effective, due to the fast saturation of its transporters [81].

Myo-Ins exhibits an active role in the InsP signaling pathway in brain tissue functionality: it plays a critical role in developing axons of sympathetic neurons; indeed, *IMPA1* transcripts encoding MIPS enzyme are highly expressed in these neurons [82,83]. Various studies report the crucial role of inositol phosphate in the CNS, revealing that alterations in its biosynthesis can affect normal brain functionality. Several works pointed out that alterations in brain myo-Ins levels are linked to severe behavioral problems and neurological deficits, including Alzheimer’s disease [84], obsessive–compulsive disorder [85,86], and autism spectrum disorder [87], as well as suicide [88] and stroke [89]. In addition, two studies on geographically isolated consanguineous families in northeastern Brazil and in a cohort in Pakistan demonstrated that a homozygous frameshift mutation in the gene coding for the enzyme IMPA1 was associated with severe intellectual disability [89,90,91].

Considering that myo-Ins can be obtained both through biosynthetic mechanisms and from dietary sources, it could be hard to conceive inositol deficiency. However, alterations in myo-Ins biosynthesis or reduced food supply often result in inositols deficiency, contributing to the development of numerous chronic diseases.

## 4. Defects in Inositol Clearance and Renal Excretion

The plasmatic concentration of myo-Ins strictly depends on the balance of dietary inositol intake, cellular uptake, endogenous synthesis from glucose, cellular metabolism, and its clearance [63,92,93,94].

The process of myo-Ins clearance consists in its catabolism and degradation to D-glucuronate, which is made up by a kidney-specific enzyme, namely myo-Ins oxygenase (MIOX), and the glucuronate-xylulose (GX) pathway, followed by renal excretion. The kidney indeed is the major site for de novo myo-Ins synthesis, reabsorption and degradation [95], working as the most important organ in regulating plasma inositol concentration in animals and humans [92]. The MIOX enzyme is selectively expressed in the renal proximal tubular compartment, and alterations in its expression levels or activity may lead to an altered degradation of myo-Ins causing a depletion in its levels, resulting in renal and metabolic complications [64,96,97].

Several studies associated renal depletion of myo-Ins with the pathogenesis of metabolic diseases, including obesity, hypertension and diabetes, and also with diabetic complications, such as diabetic nephropathy (DN) [96]. Notably, serum and urine concentration of MIOX is considered a promising novel biomarker for the early diagnosis of DN [96]. Furthermore, mouse models of insulin resistance and hypertension exhibited myo-Ins depletion as a persistent feature: the authors did not observe any changes in gene expression of *MIPS* and *IMPase*, suggesting that the inositol biosynthetic pathway was unaltered among such different conditions [96].

According to the “sorbitol-myo-Inositol hypothesis”, the depletion of myo-Ins and the impaired phosphoinositide metabolism can be considered a consequence of higher glucose levels [98]. Indeed, high glucose concentrations can induce increased activity of the polyol pathway, which is responsible for the reduction of glucose into sorbitol, significantly raising cellular osmolarity. To counteract this event, cells actively inhibit the uptake of other osmolytes, including inositols, promoting their cytosolic depletion. Myo-Ins indeed acts as an organic osmolyte that enables renal cells to constantly adapt to hyperosmotic environments.

However, further studies in diabetic rats highlighted that myo-Ins depletion can also occur independently of the polyol pathways. Higher glucose concentrations can directly upregulate MIOX enzyme activity [99], resulting in myo-Ins depletion. This regulation involves (i) polymorphisms in the promoter regions of *MIOX* gene [100], (ii) the activation of several transcription factors by different forms of stress induced by hyperglycemia [101], and (iii) post-translational modification of MIOX induced by kinases including protein kinase A and C and the 3-phosphoinositide-dependent protein kinase 1 [101].

In line with this evidence, previous studies corroborated the striking association with increased MIOX expression in kidneys from hypertensive, insulin-resistant, and 4-week diabetic animals, suggesting that myo-Ins degradation is likely a significant regulator of intracellular myo-Ins levels within the kidney [96].

Furthermore, the upregulation of MIOX expression may lead to increased levels of reactive oxygen species (ROS) exacerbating renal tubular injury in various pathological states. Myo-Ins depletion may reflect increased catabolism in the proximal tubules through MIOX activity and GX pathway, promoting oxidative stress. Indeed, studies on mouse models overexpressing the *MIOX* gene (MIOX-TG mice) indicated the concomitant activation of the glucuronate-xylulose pathway, in which myo-Ins is converted into xylulose and ribulose, along with the generation of ROS. The latter would adversely affect the pathobiology of the tubulo-interstitial compartment of the kidney; indeed MIOX-TG diabetic mice also exhibit a greater worsening of renal functions.

Mechanisms of inositol endogenous synthesis, adsorption from diet and its clearance were strictly linked to each other, and overall alterations in any of these processes could affect intracellular myo-Ins levels, reflecting renal and metabolic alterations.

## 5. Defects in Inositol Adsorption

The adsorption of all nutrients, as well as inositols, from diet into the blood is due to the highly polarized epithelial cell layer forming the intestinal mucosa. In particular, the transport proteins responsible for the adsorptive function of the gastro-intestinal tract reside in the apical side of the intestinal villous, which are involved in facilitating the nutrients transport across the small intestine [102]. Cellular inositol adsorption is mediated by passive and active processes via specific transporters; therefore, genetic alterations and/or mutations leading to reduced numbers or activity of the transporters, can induce depletion of myo-Ins content. Additionally, potential disturbances to intestinal adsorption may lead to a condition of malabsorption, causing a nutritional deficiency.

### 5.1. Inositol Transporters Alterations

The adsorption of myo-Ins or inositol-phosphate derivatives (including InsP6) from the diet occurs in the gut. At high concentrations, cellular adsorption occurs by a diffusion process, while the active uptake is primarily carried out by a system of transporters. The Na^+^-coupled transport is exerted by two higher-affinity transporters, namely Sodium/myo-Inositol Transporter-1 (SMIT1) and Sodium/myo-Inositol Transporter-2 (SMIT2), while the H^+^-coupled transport is exerted by the lower affinity H^+^/myo-Inositol transporter (HMIT). These transporters exhibit a different distribution in the human body. HMIT is primarily expressed in brain and less in kidneys, adipocytes, and oocytes [103]. SMIT1 and SMIT2 transporters belong to the *SLC5* human gene sub-family of the Na^+^-dependent glucose cotransporters [104], and their codifying genes exhibit similar expression patterns. *SLC5A3*, encoding SMIT1, is expressed in kidney, brain, placenta, pancreas, heart, skeletal muscle, and lung [100,103]. *SLC5A11*, encoding SMIT2, exhibits high expression in the small intestine, kidney, heart, skeletal muscle, liver, and placenta, and it is weakly expressed in brain [105,106].

Several studies reported that an altered genic expression is involved in different diseases and pathological processes. Alterations in the expression of *SLC5A3* gene are directly related to reduced myo-Ins content. Mouse models that are homozygous null for *SLC5A3* exhibit a 77% reduction in myo-Ins content during the embryonal development, which is still evident at late fetal stages (84% of reduction). In addition, the absence of *SLC5A3* gene product is lethal under normal conditions. A consistent study revealed that *SLC5A3*-null mice, lacking SMIT1, die shortly after birth due to neurological dysfunctions and respiratory failure [107]. Interestingly, the addition of myo-Ins to the drinking water of mother mice during pregnancy can rescue the survival and viability of the offspring [107,108,109]. However, surviving *SLC5A3* knockout mice exhibit myo-Ins depletion in brain, kidney, skeletal muscle, and liver, along with severe abnormalities in peripheral nerves, sciatic nerve and bones [106]. These pleiotropic effects underlined the importance of SMIT1 activity during fetal development and its contribution to fetal myo-Ins levels, although the mechanisms for the impact of altered SMIT1 expression are still under discussion [106].

### 5.2. Inositol Resistance: A Competition with Structurally Similar Molecules

The detection of SMIT2 RNA in the small intestine mucosa suggests that SMIT2 may be the primary transporter that mediates intestinal absorption of myo-Ins [103]. Several studies reported that the competition with structurally similar molecules may reduce myo-Ins transport and adsorption. Garzon and colleagues demonstrated that d-chiro-Ins, another natural stereoisomer, may induce an inhibitory effect on myo-Ins adsorption. They observed that when administered at high dosage (6000 mg myo-Ins with 1000 mg d-chiro-Ins), d-chiro-Ins may compete with myo-Ins adsorption, exhibiting a higher affinity selectively for SMIT2, thus interfering and inhibiting the intestinal transport of myo-Ins. This aspect should be carefully evaluated when it is necessary to achieve the correct dietary supplementation of inositols [26]. Various expert points of view revealed the importance of the appropriate use of inositols [110,111], considering that both the stereoisomers play a crucial role in controlling metabolism, hormonal signal transduction and ovarian function [112]. The administration of d-chiro-Ins only is not therapeutically useful, since the unidirectional physiological conversion of myo-Ins into d-chiro-Ins. On the contrary, myo-Ins-based administration is effective, thanks to its conversion into d-chiro-Ins. However, the combined administration of myo-Ins and a small dosage of d-chiro-Ins enhances the therapeutic effect, since d-chiro-Ins is immediately functionable. Therefore, a therapeutic combination of myo-Ins and d-chiro-Ins should reflect the physiological ratios in plasma and follicular fluid ranging from 40:1 and 100:1 [113]. Notably, myo-Ins: d-chiro-Ins 40:1 ratio is considered the first-line approach to the integrative treatment with inositols for hyperinsulinaemic PCOS patients [19], defined as the most appropriate strategy to correct metabolic aspects in PCOS patients [114].

In addition, other molecules including sorbitol and glucose are structurally similar to myo-Ins affecting its transport system. There is a competition for sodium availability of the transport processes of myo-Ins (SMIT2) and glucose (sodium-linked glucose transporter 1, SLGT1). Studies investigating similarities between SGLT1 and SMIT2, which are members of the same gene family, revealed that they interact with the same inhibitors: blocking the main sodium-glucose transporters (SGLT 1/2) prevents both glucose and inositol uptake, suggesting that the two molecules share the transporter systems [115].

In more detail, glucose significantly counteracts cellular uptake of inositol by competitive mechanisms due to the similar structure of the two molecules. A condition of hyperglycemia may counteract cellular uptake of inositol as it was observed that elevated glucose concentrations in medium reduced myo-Ins uptake in rabbit peripheral nerve tissue [116]. Evidence in vitro confirmed that 20 mM glucose concentration may significantly inhibit inositol uptake in cell cultures [117], indeed in vitro high glucose levels attenuate myo-Ins concentrating capability via competitive inhibition. Interestingly, several studies demonstrated a negative association between phytic acid intake and glycemic index of cereals and legumes, leading to the hypothesis of a correlation among inositol, phytic acid, and glucose metabolism [118]. Indeed, the removal of phytate from bean flour increases its glycemic index [119], while myo-Ins significantly inhibits duodenal glucose absorption and reduces blood glucose rise, suggesting the existence of a competitive affinity for the same transporter system [120].

Furthermore, glucose may also induce a depletion of myo-Ins through a biochemical pathway switching; for instance, in diabetic patients, inositol depletion may arise due to the activation of the polyol pathway (glucose–sorbitol pathway), whereby glucose is first converted to sorbitol by aldose reductase and then to fructose by sorbitol dehydrogenase [121].

This was confirmed by demonstrating that preincubation with aldose reductase inhibitors may partially prevent this switching [122]. Indeed, the elevated conversion of glucose into sorbitol significantly raises cellular osmolarity; therefore, to counteract this event, cells actively inhibit the uptake of other osmolytes, including inositol, promoting their cytosolic depletion by downregulating the expression of specific carriers at the transcriptional level [123].

To corroborate this, the pharmacological inhibition of aldose reductase restores myo-Ins content in impaired diabetic peripheral nerves [117], thus improving the sodium-potassium ATPase activity. Supplementation with myo-Ins can efficiently counteract such abnormalities, improving nerve conduction in diabetic animals [124,125].

Moreover, the switching toward the polyol pathway and the consequent depletion of myo-Ins can increase oxidative stress and enhance the susceptibility to oxidative tissue damage due to the deletion of the reduced glutathione [98]. This evidence highlights the reason why peripheral nerve integrity is strictly dependent on inositol-pathways [126], strengthening the link between hyperglycemia and deregulated inositol metabolism.

A compelling body of evidence corroborated the role of glucose in cellular inositol storage and depletion, highlighting that myo-Ins and glucose metabolism are interlinked and that abnormalities in metabolism of both molecules are associated [13]. Studies in hyperglycemic patients revealed an intracellular depletion of myo-Ins, especially in those tissues susceptible to developing diabetes complications [127]. In addition, hyperglycemia and insulin resistance modify the relative proportion between the two most abundant stereoisomers, d-chiro-Ins and myo-Ins [128]. Changes in the ratio of plasma and urinary myo-Ins/ d-chiro-Ins levels are so tightly linked to insulin abnormalities to be considered an early marker of hyperglycemia and insulin resistance [13]. Indeed, as previously described, myo-Ins depletion may worsen insulin resistance and diabetes complications, including redox state, free-radical defense and renal function [129,130].

Interestingly, numerous studies indicated that dietary supplementation of myo-Ins can reduce postprandial glucose levels and increase peripheral insulin sensitivity [14,64,131,132], thus improving several diabetes symptoms, as well as metabolic markers in a wide range of pathological conditions [133].

### 5.3. Microbiota Alteration and Intestinal Inflammation Reduce the Adsorption

An altered composition of gut microbiota, unbalanced towards harmful microorganisms, may induce intestinal inflammation, leading to poor adsorption of nutrients and to a nutritional deficiency.

The human intestinal microbiota is made up of trillions of microorganisms, living in a symbiotic relationship with the host by protecting against pathogens colonization and invasion. It also performs an essential metabolic function, contributing to the extraction of energy and nutrients, such as short-chain fatty acids (SCFAs) and amino acids from food [134]. A condition of altered gut microbial composition, referred to as dysbiosis, induces a reduction of SCFAs synthesis with increasing intestinal inflammation and a weakness of intestinal junctions. This promotes a condition of endotoxemia with reduced adsorption of nutrients and micronutrients, including inositols.

Low-fiber diets, genetic alterations, or some drugs can alter the composition of gut microbial populations toward harmful microorganisms, exposing patients to a higher risk of developing several chronic diseases, such as inflammatory bowel disease, type 2 diabetes, obesity and PCOS [135,136,137,138]. Furthermore, a diet low in antioxidants can induce conditions of increased oxidative stress that are associated with chronic inflammation of low intensity [139,140], leading to poor intestinal adsorption.

In addition, intestinal dysbiosis and the consequent inflammation can determine a poor effectiveness of dietary supplementations that aim to restore nutritional deficiency. For instance, previous studies on women affected by PCOS, which is characterized by an ovarian depletion of myo-Ins, revealed that dietary supplementation of inositols was not effective in all the evaluated patients, due to the problem of inositols resistance and to their poor intestinal adsorption. In this case, the concomitant use of a prebiotic, namely α-lactalbumin (α-LA), exhibited a higher effectiveness of dietary supplementation by enhancing inositols intestinal adsorption and overcoming the problem of resistance [141]. Indeed, several works reported α-LA positive effects on gut microbiota in obese mice by stimulating beneficial microorganisms, including Lactobacilli and Bifidobacteria, and by improving intestinal adsorption and its trophism, along with the metabolic profile [142].

## 6. Iatrogenic Depletion of Inositols

The chronic use of several drugs can affect gut microbiota composition, leading to a condition of dysbiosis, causing inflammation and poor intestinal adsorption. Indeed, the composition of the human gut microbiome as a complex ecosystem can be influenced in a bidirectional interaction by drugs, and conversely, gut microbiome can also influence an individual’s response, by enzymatically transforming a drug’s structure and altering its bioavailability, bioactivity or toxicity [143]. Interestingly, various commonly used non-antibiotic drugs, such as proton pump inhibitors (PPIs), statins, laxatives, metformin, betablockers and selective serotonin reuptake inhibitor antidepressants may affect microbiota composition, leading to an inflammatory state and a condition of poor adsorption of micronutrients, including inositols.

Notably, some drugs can directly target myo-Ins content by inducing a reduction of its levels according to “the inositol depletion hypothesis” [8]. In particular, drugs commonly used in the treatment of bipolar disorders and/or epilepsy aim to reduce brain inositol levels as a common therapeutic outcome. Indeed, both these pathological conditions are characterized by an over-activation of InsP3/Calcium (Ca^2+^) cerebral signaling [144] in affected patients. Myo-Ins is physiologically involved in brain functions, and its higher levels are reported in firing both manic phases of bipolar disorder and epileptic seizures. Therefore, the most used anticonvulsant drugs, valproic acid (VA) and carbamazepine (CBZ), and the mood stabilizer lithium (Li^+^), exhibit the depletion of myo-Ins in the CNS as a common mechanism of action.

In more detail, Li^+^ and VA affect myo-Ins levels by targeting enzymes involved in its synthesis and recycling. In particular, Li^+^ acts by inhibiting two phosphatases, the inositol-1,4 bisphosphate 1-phosphatase (IPP) and the inositol-1(or 4)-monophosphatase (IMPase), while VA can inhibit another enzyme, the inositol synthase (INO1) [8]. In addition, all the three drugs may inhibit SMIT1 activity, which is responsible for inositol uptake into the brain cells. However, myo-Ins biosynthesis and recycling are crucial processes for inositol brain levels since it poorly passes the BBB, thus limiting its adsorption.

Notably, all these medications exhibit a defined narrow therapeutic window, and side effects occurring during chronic treatments were mostly associated with inositol depletion in peripheral tissues. Experimental and clinical evidence reported that the chronic use of Li^+^, VA and CBZ may expose patients to peripheral side effects related to several pathological conditions, such as PCOS, hypothyroidism, hormonal, and metabolic imbalances such as weight gain, hyperinsulinemia, and dyslipidemia [145,146]. Interestingly, all these pathological conditions were associated with altered myo-Ins metabolism, such as reduced levels in related peripheral tissues. Indeed, Sherman and colleagues reported that the inositol depletion occurring in the CNS after Li^+^ administration correlates with reduced myo-Ins levels in peripheral tissues, such as the kidney and testes [147]. Numerous pieces of evidence revealed that myo-Ins depletion in renal tissue is associated with its increased degradation, due to the increased activity of MIOX enzyme in animal models of metabolic disease, such as diabetes mellitus, dietary-induced obesity, and hypertension [96]. Indeed, conditions like diabetic nephropathy are characterized by high oxidative stress, which promotes the upregulation of MIOX and the consequent depletion of myo-Ins and its isomers and phosphate derivatives. A recently published work revealed that toxic exposure to cadmium may also induce MIOX overexpression by inducing high levels of oxidative stress [148]. The authors demonstrated the efficacy of myo-Ins supplementation in determining a significant reduction of MIOX expression, along with a direct antioxidant effect, thus improving biochemical and morphological parameters in renal function [148]. Another related study reported the effective antioxidant effect of the combined myo-Ins and Selenium administration also on thyroid functionality, by observing a protective effect from cadmium-induced toxicity in a mice model [149].

Notably, myo-Ins depletion is also associated with a condition of hypothyroidism, due to its central role as a second messenger in the thyroid-stimulating hormone (TSH) pathway [150]. About 20% of patients taking lithium and anticonvulsant treatments exhibit affected thyroid functionality, with a prevalence of hypothyroidism. Furthermore, VA chronic treatment can specifically induce the occurrence of symptoms related to PCOS, including an altered endocrine and metabolic unbalance typically correlated with altered inositols metabolism in these patients. In detail, PCOS patients are generally characterized by an altered ratio between myo-Ins and d-chiro-Ins in favor of the former. These patients tend to exhibit insulin resistance, resulting in reduced intracellular conversion of myo-Ins to d-chiro-Ins [16,18,151]. An opposite situation occurs in the ovaries of PCOS patients, which maintain normal sensitivity to insulin [113,152,153], becoming enriched in d-chiro-Ins and depleted in myo-Ins. This depletion promotes hyperandrogenism and the related features (hirsutism, acne) due to the physiological roles played in the ovaries by myo-Ins as a second messenger of the follicle-stimulating hormone (FSH), and by d-chiro-Ins, which is responsible for insulin-mediated androgens synthesis [80,154].

In addition, Li^+^ administration can induce dermatological effects, including psoriasis. Interestingly, a recent work by Owczarczyk-Saczonek Agnieszka and colleagues identified an intriguing role for inositols, in particular d-chiro-Ins, as adjuvants to the local treatment of mild plaque psoriasis, opening novel applications [155]. Indeed, a recent in vitro study revealed that the high expression of aromatase enzyme, which is involved in estrogens production, and which is inhibited by d-chiro-Ins, is associated with a reduced skin elasticity [156].

Overall, drug-induced depletion of myo-Ins also influences d-chiro-Ins levels, affecting the downstream process of conversion. For this reason, a combined administration of myo-Ins and d-chiro-Ins is therapeutically more effective rather than myo-Ins only. The combined ratio of inositols may range from 10:1 to 100:1, including the ratio 80:1 in favor of myo-Ins for recovering iatrogenic inositols depletion.Finally, all the three drugs, Li^+^, VA and CBZ [157], should be avoided in pregnant women, or in fertile ones, given that these medications may expose them to teratogenicity risk regarding malformations and developmental problems of the exposed child, increasing the risk of NTDs. Notably, previous studies revealed the active role of myo-Ins in neural tube closing, suggesting its protective role during the periconceptional period of pregnancy in preventing the risk of NTDs, especially for folic acid-resistant NTDs [24,30].

## 7. Therapeutic Strategies Based on Inositols Supplementation

Dietary habits in developed countries have contributed to reduced inositol intake, due to reduced consumption of whole grains, legumes, and nuts, which are all foods enriched in the main source of inositol as phytic acid. At the same time, genetic alterations and/or unbalanced diets, along with the use of some drugs, led to inositols depletion, which was correlated to several pathological conditions.

In this regard, several studies elucidated the positive effects of myo-Ins dietary supplementation in various diseases associated with its depletion [9], including insulin resistance, PCOS [132], diabetes, gestational diabetes [24], depression [29], and metabolic syndrome. Therefore, several studies investigated this topic evaluating different forms, combinations, and dosages of inositols with variable results [9].

In clinical practice dietary supplements usually contain no more than 4 g/day of inositol, even though studies in patients with depressive disorders used much larger doses up to 12–18 g/day, without reporting significant adverse events, while showing additional clinical benefits.

Noteably, referring to the use of myo-Ins supplementation, it is crucial to emphasize the safety of its administration. The U.S. Food and Drug Administration (FDA) included myo-Ins among compounds generally recognized as safe (GRAS), implying that it is considered safe by experts, meeting the food additive tolerance requirements of the Federal Food, Drug, and Cosmetic Act (FFDCA). In addition, previous studies reported that a dosage of myo-Ins in a range of 12–30 g/daily can induce only mild gastrointestinal symptoms experienced for the first month [158,159]. Notably, the dosage of 4 g/day of inositol commonly used in clinical practice is completely free of side effects [159].

Considering the plethora of mechanisms contributing to inositol depletion, it is crucial to figure out the leading cause in order to provide the most appropriate therapeutic strategy possible.

When genetic alterations cause a reduced number of inositols transporters or a decreased activity, a tailored therapeutic approach should be organized in more than a single daily administration, avoiding in this way their fast saturations, which determines a consequent reduction in inositols levels.

In the cases of poor inositols intestinal adsorption due to glucose competition, the administration of inositol is recommended far from meals, overcoming the competitive adsorption in a condition of high glucose levels. Indeed, thanks to these competitive mechanisms, in patients affected by diabetes and insulin resistance, a dietary supplementation of myo-Ins is able to ameliorate glucose metabolism by reducing postprandial glucose levels and increasing peripheral insulin sensitivity [14,64,131,132].

A defective intestinal adsorption of inositols can be further linked to a condition of gut dysbiosis, leading to an inflammatory state and to a malabsorption of micronutrients. In these cases, the concomitant supplementation of inositols (myo-Ins and d-chiro-Ins) and prebiotics, such as α-LA, can improve intestinal eutrophism, overcoming the problem of inositol adsorption. Indeed, the use of a prebiotic like α-LA can restore the physiological gut microbiota composition in favor of beneficial microorganisms like Lactobacilli and Bifidobacteria, improving inflammatory status and metabolic profile. Previous studies demonstrated that α-LA supplementation is able to recover the altered gut microbiota composition and related inflammation in the mouse model of obesity [142]. Furthermore, studies in PCOS women revealed that the concomitant administration of myo-Ins, d-chiro-Ins, and α-LA can overcome the problem of inositols resistance by improving their intestinal adsorption, thus enhancing their beneficial effect on reproductive and metabolic profile [141,160].

Finally, a concomitant assumption of mood stabilizers and anticonvulsant drugs with a controlled dosage of inositols may recover, or altogether avoid, the occurrence of side effects during pharmacological therapy. By using a controlled dosage that poorly passes the BBB, dietary supplementation of inositol can restore peripheral levels of myo-Ins in patients taking Li^+^, VA, or CBZ, avoiding interference with their central therapeutic effect. Several works corroborated this evidence by revealing the positive effects of myo-Ins supplementation in restoring side effects related to inositol depletion-induced drugs, without interfering with the central therapeutic reduction. Inositol supplementation (3 g/daily), both in rats and in patients, ameliorated Li^+^-induced polyuria-polydipsia [161], which is one of the most common unwanted events in these patients, without any effects on Li^+^ central therapeutic outcome [161]. Furthermore, Allan and colleagues demonstrated that myo-Ins administration (6 g/daily) in patients with psoriasis under Li^+^ treatment is able to recover the Psoriasis Area and Severity Index (PASI), further helping psoriasis aggravated by Li^+^ treatment [162], without dampening the central myo-Ins depletion and without any negative effects on mood disorder. Subsequently, a case report of a bipolar patient in whom Li^+^ treatment was discontinued due to a severe psoriatic exacerbation revealed that after myo-Ins administration (3 g/daily), skin condition significantly improved, while interestingly, patient mood remained stable [163].

As previously reported, the induced depletion of myo-Ins also influences d-chiro-Ins levels, affecting the downstream process of unidirectional conversion, starting from myo-Ins. Indeed, a recent work revealed a positive effect of d-chiro-Ins topical application on psoriatic plaques. For this reason, the administration could be encouraged of a combination between myo-Ins and d-chiro-Ins rather than only myo-Ins. The combined ratio reflecting physiological levels of inositols may range from 10:1 to 100:1, especially encouraging the ratio 80:1 in favor of myo-Ins to counteract the iatrogenic depletion.

Previous studies suggest a crucial therapeutic use for inositol, up to 6 g/daily, in patients with psoriasis who need to continue to take lithium for the management of bipolar affective disorders. Noteworthy, scientific evidence reported that myo-Ins is poorly absorbed from the periphery into the brain, and as a result, when administered exogenously, large doses are required for penetration into the CNS [164]. Indeed, previous studies reported beneficial effects of myo-Ins supplementation in depressive patients and in premenstrual dysphoric disorder by using a higher dosage of 12 g/daily to cross the BBB [29]. Therefore, considering the positive effects of myo-Ins in recovering side peripheral effects, its supplementation in a controlled dosage that poorly passes the BBB (up to 6 g/daily), can effectively recover the adverse effects without hindering the beneficial central action of the pharmacological treatment.

Overall, considering the safety of inositols administration and their wide use in various pathological conditions previously described, it is intriguing to propose a diversified inositols supplementation depending on the mechanisms of depletion, with the aim to provide a therapeutic approach as tailored as possible.

## 8. Conclusions

Cellular and tissue homeostasis of inositols depends on several mechanisms including its endogenous synthesis and catabolism, transmembrane transport, intestinal adsorption, and renal excretion.

Alterations in these mechanisms, due to genetic alterations, competitive processes, the use of some specific drugs aiming to reduce myo-Ins levels in the CNS, can lead to a reduction of inositol levels correlated to several pathological conditions. Indeed, myo-Ins as a key player in various signal pathways is involved in different physiological processes, including reproductive, hormonal, and metabolic modulation. Altered inositols levels or metabolism were correlated to several disorders, such as PCOS, hypothyroidism, hormonal, and metabolic imbalances, like weight gain, hyperinsulinemia, and dyslipidemia.

A deeper knowledge of the mechanisms involved in inositols depletion (Table 1) can add new insights to shape future therapeutic approaches as tailored as possible. It is crucially important that dietary supplementation of inositols takes into account the different mechanisms determining inositols depletion, shaping the dosages and the route of administration, with the aim to develop an efficacious and safe approach in order to counteract inositols depletion-induced pathological events.

## Figures and Tables

**Table 1 ijms-22-06796-t001:** Mechanisms involved in inositols depletion.

	Mechanisms of Inositols Depletion
Reduced nutritional availability	Highly refined processes eliminating phytate from food
	Social and educational different habits among countries
Defects in biosynthesis	Epigenetic modulations (methylation) of *ISNYA1* gene
	GSK3 positive modulation of *ISNYA1* gene
Defects in clearance and renal excretion	Up regulation of MIOX enzyme activity
Defects in inositols adsorption	Alterations in Inositol transporter (SMIT1)
	Inositol resistance: competitive mechanisms due to structurally similar molecules (such as d-Chiro-Ins, Glucose, etc.)
	Altered microbiota composition (dysbiosis) and intestinal inflammation
Iatrogenic depletion	Drugs affecting microbiota composition
	Drugs reducing myo-Inositol levels in the brain and in peripheral tissues (Lithium, Valproic Acid, Carbamazepine)

The table summarizes the main mechanisms involved in inositol depletion.

## Data Availability

Not applicable.

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
