# Peer review of "Inositols Depletion and Resistance: Principal Mechanisms and Therapeutic Strategies"

_ijms, 2021, doi:10.3390/ijms22136796_

Round 1

Reviewer 1 Report

In the present review “Inositols depletion and resistance: principal mechanisms and therapeutic strategies”, Lepore and coworkers examine the principal mechanisms causing Inositols depletion and resistance paving the way for tailored therapeutic strategies based on inositol supplementation. Overall, I think that the manuscript is well-written (within the scope of this journal), well-structured and the data are of clinical relevance on a current topic of interest.

I have some suggestions to improve the quality of review.

1) Please discuss the following papers in paragraph n.6 of the manuscript:

  • Benvenga et al. Nutrients. 2020 Apr 26;12(5):1222.
  • Pallio et al. Food Chem Toxicol. 2019 Oct; 132:110675.

As a matter of fact, these recent papers add new findings on mechanism of action of Myo-Ins (Thyroid and Kidney in a murine animal model) and on therapeutic strategies based on inositol supplementation.

2) Please check the correct format: lines 221-255; References n°5, 6, 29, 38, 42, 66, 98, 160.

Author Response

Point 1. Please discuss the following papers in paragraph n.6 of the manuscript:

  • Benvenga et al. Nutrients. 2020 Apr 26;12(5):1222.
  • Pallio et al. Food Chem Toxicol. 2019 Oct; 132:110675.

As a matter of fact, these recent papers add new findings on mechanism of action of Myo-Ins (Thyroid and Kidney in a murine animal model) and on therapeutic strategies based on inositol supplementation.

Response 1. We thank the Reviewer for his/her endorsement about the submitted manuscript and for all the suggestions that we carefully took into consideration. We added some explanations regarding the indicated studies, as you can see at lines 518-527 in the resubmitted manuscript. We expanded the discussion reporting evidence about the effectiveness of Myo-Inositol administration in recovering renal and thyroid functionality in a condition of oxidative damage induced by cadmium. Indeed, these papers add new insights on Myo-Ins positive role in recovering thyroid and renal altered functionality.

Point 2. Please check the correct format: lines 221-255; References n°5, 6, 29, 38, 42, 66, 98, 160.

Response 2. We checked all the minor flaws throughout the text correcting all the indicated references and lines. We deleted sentences reported at lines 221-222, they were not to be present, and we correct the format at lines 223-255.

Reviewer 2 Report

Submitted for review, this Review essentially deals with the causes and problems associated with the deficiency of inositols in the body. The authors essentially refer to myo-inositol as it is the most popular natural steroisomer.
The review is well done and divided into well organized chapters. The references are sufficient and updated. The final table is sufficient to give a summary picture. I have no particular suggestions other than to point out small typographical errors.

L148 608mg add space

L221-222: check the position of the sentences

L316 Add reference

L616 3grams/daily Add space

Table 1 Reduced nutritional availability check the line of the table, is too long

Author Response

Submitted for review, this Review essentially deals with the causes and problems associated with the deficiency of inositols in the body. The authors essentially refer to myo-inositol as it is the most popular natural steroisomer.
The review is well done and divided into well organized chapters. The references are sufficient and updated. The final table is sufficient to give a summary picture. I have no particular suggestions other than to point out small typographical errors.

L148 608mg add space

L221-222: check the position of the sentences

L316 Add reference

L616 3grams/daily Add space 627

Table 1 Reduced nutritional availability check the line of the table, is too long

Response. We thank the Reviewer for his/her endorsement about the submitted manuscript and for all the suggestions that we carefully took into consideration. We checked all the minor flows indicated by the Reviewer and we corrected all the typographical errors.

We added the missing space, as you can see at lines 148 and 628 in the new version of the submitted manuscript. We deleted the sentences at lines 221-222, they were not to be present. We further added the required reference at line 316 and we corrected the line of the Table1 so that all the lines exhibit the same length.